# Does Sexual Function and Quality of Life Improve after Medical Therapy in Women with Endometriosis? A Single-Institution Retrospective Analysis

**DOI:** 10.3390/jpm13121646

**Published:** 2023-11-25

**Authors:** Anna Biasioli, Silvia Zermano, Francesca Previtera, Martina Arcieri, Monica Della Martina, Diego Raimondo, Antonio Raffone, Stefano Restaino, Giuseppe Vizzielli, Lorenza Driul

**Affiliations:** 1Department of Maternal and Child Health, “Santa Maria della Misericordia” University Hospital, Azienda Sanitaria Universitaria Friuli Centrale (ASUFC), 33100 Udine, Italy; martina.arcieri@asufc.sanita.fvg.it (M.A.); monica.dellamartina@asufc.sanita.fvg.it (M.D.M.); stefano.restaino@asufc.sanita.fvg.it (S.R.); lorenza.driul@uniud.it (L.D.); 2Department of Medicine (DAME), Università degli Studi di Udine, Via delle Scienze, 206, 33100 Udine, Italy; previtera.francesca@spes.uniud.it; 3Division of Gynecology and Human Reproduction Physiopathology, IRCCS Azienda Ospedaliero-Universitaria di Bologna, 40133 Bologna, Italy

**Keywords:** endometriosis, quality of life, medical therapy, chronic pain, central sensibilization phenomena, multidisciplinary approach

## Abstract

Background: Endometriosis is a gynecological condition affecting up to 10% of women of reproductive age and characterized by chronic pain. Pain is the major cause of the impairment of quality of life in all aspects of these patients. Previous studies have shown that endometriosis treatment, hormonal or surgical, has proven effective not only in controlling the disease but also in improving symptoms, and we can assume also effective in improving quality of life. Methods: This study evaluates quality of life and sexual function in patients with endometriosis at the time of diagnosis and after 6 months of medical therapy, to assess the impact of treatment on these aspects. We evaluated retrospectively patients with a diagnosis of endometriosis between 2018 and 2020. All patients underwent gynecological examination and transvaginal ultrasound and filled in three questionnaires. The same evaluation was provided after taking medical hormonal therapy. Results: The improvement of dysmenorrhea, chronic pelvic pain, and dyspareunia after medical treatment were statistically significant. Instead, items concerning arousal, lubrication, and sexual satisfaction showed a statistically significant worsening after therapy. Conclusions: We can state that hormone therapy alone is not sufficient to achieve an improvement in the patient’s quality of life and sexual function. Emerging evidence suggests that most of these patients showed a central sensibilization phenomenon characterized by an amplification of the response to a peripheral and/or neuropathic nociceptive trigger, which is expressed by hyperalgesia and allodynia. For this reason, in these patients, it is better to adopt a multimodal and multidisciplinary approach, including other professional figures, that acts on pain and also intervenes in all those conditions that contribute to worsening quality of life.

## 1. Introduction

Endometriosis is a benign gynecological condition characterized by the presence of endometrium-like epithelium and/or stroma in ectopic locations [1].

However, this definition does not describe the complex symptomatic, pathogenetic, biological, and multisystem nature of the disorder.

The postulated origins of endometriotic tissue are retrograde menstruation, coelomic metaplasia, and lymphatic and vascular metastasis [2].

Ectopic endometriotic tissue causes dysfunction of the innate and adaptive immune system, causing a localized immune and inflammatory response, with the production of cytokines, chemokines, and prostaglandins [3].

The presence of ectopic endometrial tissue triggers a chronic estrogen-dependent inflammatory reaction. Pain results from increased prostaglandins, compression, and/or infiltration of adjacent nerves and can initially be understood as nociceptive inflammatory pain. Over time, increased nerve growth factor expression, increased nerve fiber density, and angiogenesis stimulate peripheral nerve sensitization.

These factors activate visceral and peritoneal nerve fibers, leading to increased pain sensitivity. As a result, endometriotic lesions develop pain mechanisms that can be activated regardless of hormonal stimuli [4].

Endometriosis is a common disease in young females, affecting up to 10% of women of reproductive age and up to 50% of infertile women [2].

Endometriosis can cause chronic pelvic pain, dysmenorrhea, dyspareunia, dyschezia, dysuria, rectal bleeding or hematuria, chronic fatigue, and infertility. These symptoms can have a significant impact on patients’ lives not only physically but also psychologically with a significant impairment of quality of life in all aspects, including sexual function, work, and social relationships [5,6].

Pain in many cases causes deterioration in sleep quality, increased stress, and reduced activity levels and increases the incidence of psychological comorbidities such as anxiety and depression. In addition, if the patient has dyspareunia, pain can impair sexual activity, causing negative consequences on psychological health, quality of a couple’s life, and intimate relationships.

Chronic pain has a strong impact on all types of social relationships leading to social isolation as the ultimate consequence. Symptoms such as chronic fatigue, mood swings, and heavy bleeding often lead women to absenteeism or the need to reduce work hours, leading some women to feel guilty. Absenteeism from work affects not only the individual and her family but also costs for the entire country. Endometriosis is a condition that impacts all aspects of life, with economic implications at both individual and community levels [7,8].

At the beginning of the 1990s, the World Health Organization (WHO) developed a project in order to create a cross-cultural instrument of quality-of-life assessment: the predominant World Health Organization Quality of Life (WHOQOL).

The concept of Quality of Life (QOL) has gained significant recognition and importance in the fields of health and medicine. Historically, biomedical outcomes took precedence over QOL in medical and health research. However, in recent decades, there has been a growing emphasis on assessing and addressing patients’ QOL, leading to a rise in the utilization of QOL evaluations [9]. Understanding QOL plays a vital role in enhancing symptom management, patient care, and rehabilitation. Patient-reported QOL can unveil issues that necessitate adjustments and enhancements in treatment and care or even reveal that certain therapies provide limited benefits. Additionally, QOL assessments help identify a spectrum of problems that may affect patients. QOL also holds a pivotal place in medical decision-making, as it serves as a predictor of treatment outcomes and thus carries prognostic significance. Furthermore, the term ‘health-related quality of life’ (HRQOL) is often defined as follows: A term that pertains to health-related aspects of QOL, typically reflecting the impact of both the disease and its treatment on disability and daily functioning. It is also recognized as the influence of perceived health on an individual’s capacity to lead a fulfilling life. More precisely, HRQOL serves as a metric for assessing the value attributed to the duration of life, adjusted for impairments, functional states, perceptions, and opportunities, all of which can be influenced by disease, injury, treatment, and health policy [10].

Since that time, several studies have investigated the role of endometriosis in decreasing HRQoL [11], sexual function, and quality of relationships with partners [12].

But at what point are we currently able to improve the quality of life of patients suffering from endometriosis with conventional therapies? Is there uniformity in measuring it, and what does it include?

In the patient with endometriosis, the evaluation of QoL can come to understand the important field of sexual functionality, often compromised for anatomical, biological, and psychological reasons, and the field of anxiety disorder, a small fragment within psychological disorders.

The treatment options for endometriosis include hormonal therapies, to achieve a hypo-estrogenic status (oral contraceptives, progestins, Danazol, GnRH agonists), pain-relieving agents (nonsteroidal anti-inflammatories, opioids), or surgical removal of endometriotic implants [13].

In choosing treatment for endometriosis, it is essential to take into account the patient’s symptoms, preferences, and age, as well as the side effect profile, the extent and location of the disease, previous treatments, and costs. Management of endometriosis (particularly, the disease involving extra-pelvic structures and cases where pain conditions overlap) requires multidisciplinary expertise. In addition, a better understanding of cross-sectional and central sensitization in endometriosis, as well as clinical differentiation between pain features, will move management away from a primarily lesion-based approach and will provide a broader spectrum of therapeutic targets [14].

In this regard, previous studies have shown that initiating treatment, either by hormone therapy or surgery, has proven effective not only in controlling the disease but also in improving symptoms and, in some cases, even resolving them [15].

The 2019 review by Grandi et al. concluded that combined oral contraceptives and progestin-only contraceptives result in a statistically significant reduction in endometriosis-related pain, resulting in improved quality of life [16].

Andres and colleagues in their review showed that Dienogest 2 mg/day is an effective therapy for endometriosis, with superior results to placebo in symptom control. The same results as GnRH analogues such as buserelin, leuprolide acetate, and triptorelin in the control of pelvic pain and other symptoms related to endometriosis have been demonstrated by this molecule. In addition, Dienogest therapy has side effects, such as abnormal vaginal bleeding, headache, breast pain, and weight gain, which are reduced compared to those caused by GnRH analogues, making it a therapy better tolerated by patients [15].

Based on these assumptions, we can assume that patients undergoing treatment, benefiting from an improvement in symptoms, may go on to experience an overall improvement in quality of life [17,18].

However, no single treatment is ideal for all patients, and the management approach chosen should be directed to the individual needs of each patient, but the scarcity of clinical data also hampers the selection of one progestin over another.

## 2. Materials and Methods

### 2.1. Study Design

The aim of this study is to evaluate the quality of life and sexual function in patients with endometriosis referred to the OB/GYN department of the Santa Maria della Misericordia Hospital in Udine, at the time of diagnosis and after 6 months of medical therapy, in order to assess the impact of progestin treatment on these aspects.

We evaluated retrospectively patients with a diagnosis of endometriosis attending the “Endometriosis and Pelvic Pain” specialized center of the Obstetrics and Gynecology Clinic in Udine, between 2018 and 2020.

### 2.2. Diagnosis of Endometriosis

Endometriosis was diagnosed in these patients by ultrasound imaging [19], magnetic resonance [20,21], or surgery [22].

### 2.3. Patient Characteristics

Every 18- to 50-year-old woman with a previous diagnosis of endometriosis was invited to fill out the questionnaires (SF-36—Short Form-36 Health Survey, STAI—State-Trait Anxiety Inventory, FSFI—Female Sexual Function Index) on paper at the clinic during the first visit, as an integral part of the medical history assessment. All patients were evaluated by the same team, consisting of a trained medical doctor and two residents. Eligible for the study were symptomatic patients, of childbearing age, wishing to undertake Estrogen–Progestogen/Progestogen therapy. The exclusion criteria were post-menopausal status, allergy/hypersensitivity to hormonal therapies, contraindications to hormonal therapy, and the patient’s refusal or desire for pregnancy. For every patient, we analyzed exhaustively endometriosis symptoms: dysmenorrhea, chronic pelvic pain, dyspareunia, defecation, urinary disorders, and the presence of limitations in carrying out normal daily activities.

All women were previously informed of the basic goals of the study, and they were briefed about the confidentiality and anonymity of the data collected. Their participation was voluntary. Informed written consent was obtained from each woman who met the inclusion/exclusion criteria.

At enrollment and the follow-up visit after 6 months, all patients underwent gynecological examination and transvaginal ultrasound (TVS) and were administered the following questionnaires:-SF-36 (Short Form-36 Health Survey) assessing quality of life and health satisfaction. It is composed of 36 multiple choice questions aggregated into 8 scales that investigate Physical Activity, Role and Physical Health, Physical Pain, Health in General, Vitality, Social Aactivities, Mental Health, Role and Emotional State; there is also a question about the change in health status during the last year [23,24]. The higher the score, the better the subject’s health.-STAI (State-Trait Anxiety Inventory), a psychological questionnaire based on a 4-point Likert scale. Low scores indicate a mild form of anxiety, while intermediate scores indicate a moderate form of anxiety and high scores indicate a severe form of anxiety [25]. -FSFI (Female Sexual Function Index), a validated tool for assessing the main aspects of female sexual dysfunction and sexual satisfaction. The FSFI contains six domains: desire, arousal, lubrication, orgasm, satisfaction, and pain. These domains assess sexual function over the past 4 weeks [26].

Among the selected patients, some had taken hormone therapy in the past or had undergone surgery, for diagnostic-therapeutic purposes. However, at the first visit, none were taking hormone therapy, and none had undergone surgery during the follow-up period. All selected patients started hormone therapy as they were symptomatic at the time of the first visit. The data were collected by trained personnel in a password-protected online chart and subsequently reviewed by the principal investigator.

### 2.4. Ethics Committee

Our departmental institutional review board approved the study (Institutional Review Board (IRB-DMIF), Department of Medicine (DAME), University of Udine. Approval Code: RIF.Prot. IRB: 069/2021). The study protocol conformed to the ethical guidelines of the 1975 Helsinki Declaration.

### 2.5. Study Power

The study was powered for a difference between the two groups equal to or greater than the mean of the SD of the two groups. The sample size was calculated according to the study design by Simon [27], using an α-error of 0.05 and a β-error of 0.90.

### 2.6. Statistical Analysis

The statistical analysis was performed using paired Student’s t test to compare the demographic, clinical data, and the values obtained at baseline with those of both follow-ups. For comparisons of the values obtained from the FSFI items, the Wilcoxon rank-sum test for analysis of non-parametric paired samples was used. The result was statistically significant when *p* < 0.05. Statistical analysis was performed using the statistical package StatView 5.01 (SAS Institute Inc., Cary, NC, USA).

## 3. Results

We enrolled a total of 54 patients with clinical, ultrasound, or histological diagnosis of endometriosis. Superficial endometriosis was present in 9 out of 54 patients (16%), ovarian endometriosis in 32 (59.2%), and deep endometriosis in 25 (46.3%). In total, 50% of patients had adenomyosis.

The mean age of the patients was 34.5 years (±8.75), and the mean BMI was 22 (20–25). The characteristics of the patients are summarized in Table 1.

One-third of our population had comorbidities, such as autoimmune pathologies, fibromyalgia, migraine, irritable bowel syndrome, and inflammatory bowel disease (Table 1).

Individualized hormone therapy was prescribed to each patient, as described in Figure 1, and was scheduled for a post-treatment follow-up visit.

As described in Table 2, at the baseline evaluation, among the population, an average NRS (Numerical rating scale) value is 7 for dysmenorrhea.

The improvement of this symptom was statistically significant as shown in Figure 2.

Chronic pelvic pain and dyspareunia also showed a statistically significant reduction after therapy as seen in Figure 3 and Figure 4.

The analysis of the questionnaires highlighted that there was no statistically significant difference regarding the improvement of the score of the SF36 questionnaire and each of the related items. There was no statistically significant difference from T0 to T6 in STAI values. Instead, there was a statistically significant difference from baseline in the FSFI score, as illustrated in Figure 5.

In particular, as illustrated in Table 3, half of the items concerning arousal (items 4 and 5) showed a statistically significant worsening after therapy, as well as item 7 concerning lubrication, item 11 concerning orgasm, and item 13 concerning sexual satisfaction.

## 4. Discussion

Our results show that there is a statistically significant improvement in symptom burden at six months after the start of hormonal treatment. This result is in line with the literature showing that all hormonal medical treatments lead to a clinically significant reduction in pain [26,28].

However, according to our data, no statistically significant difference was found in the SF-36 score level [29,30,31], nor when considering the individual items of this questionnaire (Physical Activity, Role and Physical Health, Physical Pain, Health in General, Vitality, Social Activities, Mental Health, Role and Emotional State). This result differs from other similar studies. In their systematic review, Sima et al. reported that the pre- and post-surgery questionnaire SF-36 in all the studies analyzed showed improvement in the physical functioning parameter after the interventions.

Only two recent studies have shown that QoL and sexuality, particularly libido and pain, improve after 6 months of Dienogest treatment in women with endometriosis [29,30].

Caruso and colleagues in 2015 evaluated the efficacy of hormonal treatment at 3 and 6 months follow-up. They demonstrated that the study group experienced an improvement in pain and QoL at the 1st follow-up and an improvement in sexual life at the 2nd follow-up compared with the control group. The progressive reduction in pain reported by women over the treatment period contributed to further improvement in their QoL and their sexual lives. In their study, Caruso et al. showed an improvement in some categories of the SF-36 during the 1st follow-up, mainly those related to the physical aspects. However, at the 2nd follow-up, all categories improved. In addition, they found that the frequency of sexual activity improved at the 2nd follow-up. This could be mainly due to the reduction in dyspareunia and pelvic pain [29].

Furthermore, Caruso and colleagues in 2019 confirmed the positive effects of the drug on pain and showed that QoL and sexual function continue to improve even after 6 months and 12 months of therapy; furthermore, the improvements are stable until 24 months [31].

Similarly, Mabrouk et al. reported a significant improvement in every scale of the SF-36 six months after surgery [32,33]. Despite the results, the studies only analyzed surgical therapy.

The study of La Rosa et al., instead, provided a general overview of the psychological and social impact of endometriosis and underlined the beneficial effects of different therapeutic options, including hormonal therapy, on quality of life and general well-being [8].

Caruso et al. in 2015 [29] reported an improvement in quality of life after a 6-month Dienogest therapy, with a progressive reduction of the pain syndrome reported by women over the treatment period. The Mabrouk et al. study is instead in agreement with our results and found no statistically significant differences between before and after the start of hormone therapy. In their retrospective study of 106 women, VAS scores for dysmenorrhea, dyspareunia, chronic pelvic pain, and dyschezia did not vary significantly during the preoperative period in hormonal therapy users; furthermore, SF-36 total score did not vary significantly during the preoperative period in either the COC user group or the non-users group. However, in our opinion, they used too few assessment tools to analyze these aspects (VAS and SF-36 questionnaire). In addition, another difference from our study is that patients were evaluated with different timelines depending on the surgery schedule [34].

Our data also showed no statistically significant difference related to an improvement in the STAI score [25], which assesses the patient’s state of anxiety, between before and after the start of therapy. 

Cross-sectional studies have identified elevated risks associated with the diagnosis of depression, generalized anxiety disorder, and post-traumatic stress disorder in individuals with endometriosis. Earlier reviews have demonstrated that endometriosis adversely impacts psychosocial well-being and the overall quality of life (QoL) of affected individuals. Nevertheless, the uncertain pathogenesis of endometriosis includes the origin of its psychological symptoms, a subject that remains incompletely elucidated [35]. In a systematic review that included 5419 women with endometriosis, Kalfas M et al. highlighted the role of psychosocial factors in Endometriosis, associated with pain and health-related quality of life (HRQoL). They included anxiety, depression, and catastrophising, essential target in the treatment of women with endometriosis [36]

An analysis of the FSFI questionnaire [23], an instrument to assess female sexual function and sexual satisfaction, revealed a worsening, particularly in some domains such as arousal, lubrication, orgasm, and overall satisfaction. This result might depend on the effect of hormone therapy on libido and lubrification. In 2016, Pluchino stated, in his review, that dyspareunia is not the only determinant of sexual health in these women. Indeed, endometriosis negatively affects different domains of sexual function. New, more complex aspects must be considered such as hypertonicity of the pelvic floor, chronic pelvic pain, vulvodynia, and the presence of physical and mental comorbidities that can affect sexual function, as well as personality traits and women’s expectations [37].

Our study’s limitations, which may have affected this result, are the relatively small sample size and the lack of long-term follow-up. For these reasons, the results must be interpreted with caution; however, in light of the results obtained, we can state that hormone therapy alone, which, however, determines the resolution of pain, was not sufficient to achieve an overall improvement in the patient’s quality of life and sexual function.

In recent years, molecular phenotypes have emerged among women with endometriosis that may be related to the presence of a range of diseases, such as inflammatory bowel disease, fibromyalgia, autoimmune diseases, and vulvodynia [38].

Regarding this, we have to highlight the high prevalence of comorbidities in the patients in our sample. One-third of the patients in our study are affected by pathologies frequently associated with endometriosis that could worsen the pain symptoms.

Furthermore, emerging evidence suggests that neuroinflammation could contribute to the pathophysiology of chronic pain conditions that coexist with endometriosis. Their symptoms can be caused by the dysregulation of sensory, inflammatory, and psychological domains. In particular, repeated exposures to injury, inflammation, and stress in genetically predisposed individuals result in the transition from acute pain to chronic pain, in part through neuroinflammation [39,40].

The term “Nociplastic Pain” was proposed as a mechanistic descriptor for chronic pain states not characterized by activation of nociceptors or neuropathy, but in which the clinical and psychophysical findings suggest altered nociceptive function [41]; this terminology can be applied to a diverse range of clinical conditions that share common neurophysiological mechanisms, involving various organ systems [40,42].

In particular, most of these patients with overlapping syndromes showed the “Central Sensitization Phenomena” characterized by an amplification of the response to a peripheral and/or neuropathic nociceptive trigger that is expressed by hyperalgesia and allodynia [32,34]. Furthermore, chronic pain is commonly associated with depression, anxiety, and sleep disorders that could worsen the quality of life of these women [43,44].

Some of these overlapping conditions are known as Central Sensitivity Syndromes (CSSs), a group of diseases characterized by chronic nociplastic pain such as Irritable Bowel Syndrome (IBS), Chronic Fatigue Syndrome, Restless Leg Syndrome, Fibromyalgia, Temporomandibular Joint Disorder, Migraine or Tension Headaches, Multiple Chemical Sensitivities, and Neck Injury, which can contribute to complicating the condition of the patient with endometriosis [38,45].

Furthermore, central changes may be the reason for medical/surgical treatment failure or pain recurrence in patients with endometriosis [45].

Therefore, in such a complicated situation, this kind of patient may require different treatments in order to manage the disease and improve their quality of life.

## 5. Conclusions

Recent papers highlight the possibility of integrating the evaluation of “-omics” data in improving the diagnosis of endometriosis. “Omics” are various disciplines in biology whose names end in the suffix “-omics”, such as genomics, proteomics, metabolomics, metagenomics, phenomics, and transcriptomics [44,46].

In this regard, in our opinion, this kind of information could help in predicting the individual patient’s sensitivity to different medical therapies, allowing the knowledge of precision medicine to be exploited in the development and implementation of a treatment plan for endometriosis in the future.

Our interpretation of the results leads us to state it is clear that in these women, the classic hormonal/surgical therapeutic approach cannot be sufficient.

Therefore, clinicians should probably adopt different therapeutic strategies to treat women with endometriosis and could adopt a multimodal and multidisciplinary approach, including when required, other professional figures, which, on the one hand. goes to act on the painful symptomatology, enhancing the action of conventional drug therapy, and on the other hand, intervenes in all those conditions that contribute to worsening quality of life, such as anxiety, depression, and sexual dysfunction [47].

In our opinion, a holistic patient rehabilitation program for endometriosis should cover the physical, psychological, and occupational aspects of the disease. 

This is a retrospective study to investigate the quality of life and sexual function in patients with endometriosis. Further prospective studies are needed to confirm our results in larger samples; currently, the prospective extension phase is ongoing at our institution.

## Figures and Tables

**Figure 1 jpm-13-01646-f001:**
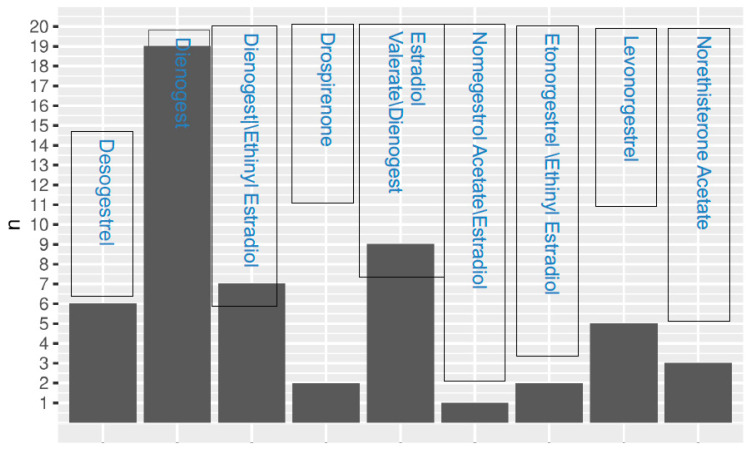
Patients’ hormonal therapy.

**Figure 2 jpm-13-01646-f002:**
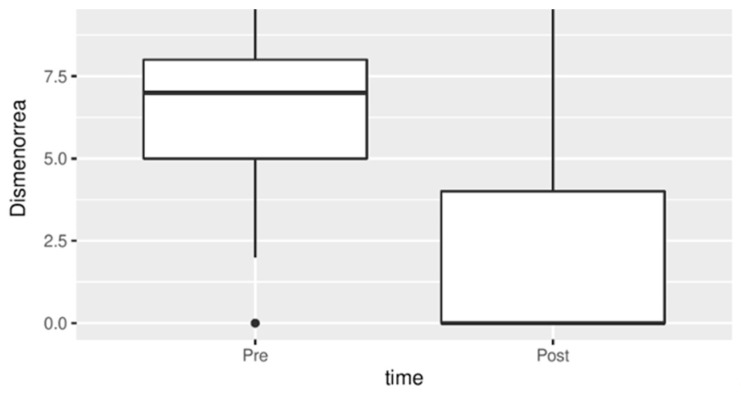
Dysmenorrhea before and after treatment.

**Figure 3 jpm-13-01646-f003:**
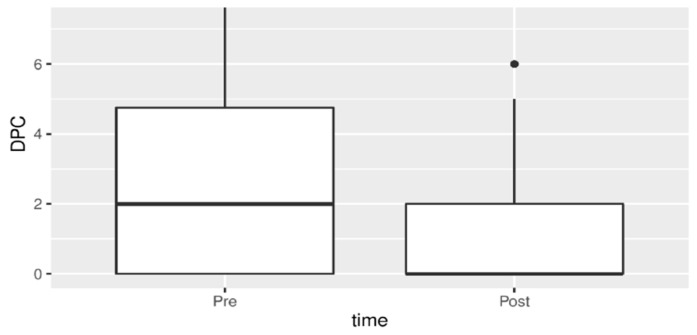
Chronic pelvic pain before and after treatment.

**Figure 4 jpm-13-01646-f004:**
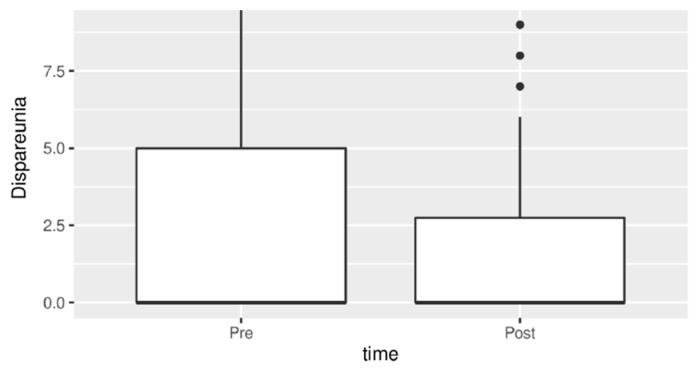
Dyspareunia before and after treatment.

**Figure 5 jpm-13-01646-f005:**
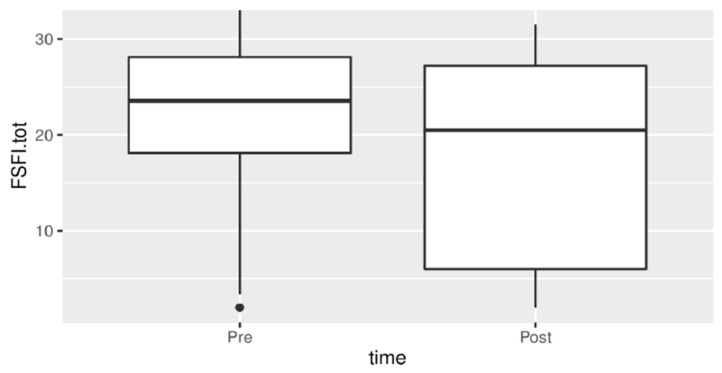
FSFI score before and after treatment.

**Table 1 jpm-13-01646-t001:** Patient characteristics (mean range).

Age	34.54 (±8.75)
BMI (Body Mass Index)	22 (20–25)
Comorbidity *	33.33% (18/54)
Smoking	25.93% (14/54)
Adenomyosis	50% (27/54)
Superficial endometriosis	16.67% (9/54)
Ovarian endometriosis	59.26% (32/54)
Deep endometriosis	46.3% (25/54)
Before therapy	
Dysmenorrhea.NRS **	7 (5–8)
Chronic pelvic pain.NRS **	2 (0–5)
Dischezia.NRS **	0 (0–0)
Dysuria.NRS **	0 (0–0)
Dyspareunia.NRS **	0 (0–5)
Post therapy	
Dysmenorrhea.NRS.1 **	0 (0–4)
Hronic pelvic pain.NRS.1 **	0 (0–2)
Discgezia.NRS.1 **	0 (0–0)
Dysuria.NRS.1 **	0 (0–0)
Dyspareunia.NRS.1 **	0 (0–3)
Before therapy	
P.STAI	45 (37–55)
P.FSFI	24 (18–28)
P.SF36	96 (85–105)
Post therapy	
D.STAI	42 (37–54)
D.FSFI	20 (6–27)
D.SF36	96 (88–108)

* Comorbidity: diseases frequently associated with endometriosis, such as autoimmune diseases (connective tissue disease, celiac disease, and thyroid disease), fibromyalgia, migraine, irritable bowel syndrome, and inflammatory bowel disease. ** NRS: Numerical rating scale.

**Table 2 jpm-13-01646-t002:** Symptoms (mean-standard deviation).

	Before		After	
	NRS	ds	NRS	ds
**Dysmenorhea**	7	(5–8)	0	(0–4)
**Chronic pelvic pain**	2	(0–5)	0	(0–2)

**Table 3 jpm-13-01646-t003:** Items FSFI score. The results highlighted in blue were statistically significant (*p* < 0.05).

ITEM FSI	CLASS	T0	T4	*p*-Value
		Mean + SD	Mean + SD	
1	DESIRE	3 (2–4)	3 (1.25–3)	0.061
2		3 (2–3.75)	3 (2–3)	0.079
3	STIMULATION	3 (1.25–5)	3 (1–4.75)	0.1
4		3 (2–4)	3 (1–4)	0.0041
5		3.5 (2–4)	2.5 (1–4)	0.011
6		4 (1.25–4.75)	3 (0–5)	0.099
7	LUBRICATION	4 (2–5)	3 (0–4)	0.016
8		2 (1–2)	1 (0–2)	0.095
9		4 (2–5)	3 (0–5)	0.062
10		2 (1–2)	1.5 (0–2)	0.13
11	ORGASM	4 (2–5)	3 (0–5)	0.023
12		2 (1–2.75)	1 (0–2)	0.084
13	SATISFACTION	4 (1.25–4.75)	2 (0–4)	0.0085
14		4 (1–5)	3 (0–5)	0.12
15		4 (3–4.75)	3.5 (3–4.75)	0.54
16		4 (2.25–4)	3.5 (3–4)	0.47
17	DYSPAREUNIA	2.5 (1–5)	3 (0–4)	0.51
18		3 (1–4)	3 (0–4)	0.93
19		3 (2–4)	3 (0–4)	0.3
TOTAL		24 (18–28)	20 (6–27)	0.036

## Data Availability

The datasets used and/or analysed during the current study are available from the corresponding author upon reasonable request.

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
