# Peer review of "Does Sexual Function and Quality of Life Improve after Medical Therapy in Women with Endometriosis? A Single-Institution Retrospective Analysis"

_jpm, 2023, doi:10.3390/jpm13121646_

Round 1
Reviewer 1 Report
Comments and Suggestions for Authors
Endometriosis studies are of persistent interest and as long we do not have greater characterization for the lesions and treat more specifically there will be a continuous strive for improving patients' life. This study does a great job analyzing sexual function and quality of life after medical therapy in women with endometriosis. It, also, highlights the need for multidisciplinary approach and greater emphasize on the psychological support. But there are limitations, acknowledged by the authors, concerning the number of cases and further follow-up. In addition I think that including data on the medical treatment would have been shedding light on possible effects upon sexual function following treatment.
Comments on the Quality of English Language
On the English language, the wording is fine with the exception of the word 'lubrification' which is less used instead of 'lubrication'. Also, table 3 is not translated from italian.
Author Response
Thank you very much for your comment on our work. I have made the requested adjustments and you will find highlighted in the uploaded revised version.

Reviewer 2 Report
Comments and Suggestions for Authors
Reviewer Comments
Page 1 – Introduction – Paragraph 3 – Line 44 – “As a consequence of infiltration of anatomic structures, correlated inflammatory reaction, nociceptive pathway,” This statement is awkward as written. Also, further describe/define “nociceptive pathway”. The authors should revise accordingly.
Page 2 – Introduction – Paragraph 7 – Lines 69-71 – “Furthermore, Caruso and colleagues in 2019 confirmed that this benefit persists in prolonged therapies.[14]” This statement is unclear. Further explain “prolonged therapies”. The authors should revise accordingly.
Page 2 – Materials and Methods – Paragraph 1 – Lines 80-82 - “We evaluated retrospectively patients with diagnosis of endometriosis attending to "Endometriosis and Pelvic Pain" specialized center of the Obstetrics and Gynecology Clinic in Udine, between 2018 and 2020, with a systematic sampling.” Describe who collected the data (investigator versus trained personnel), the records used (EMR versus paper charts) and define the term “systematic sampling”. It’s assumed that a retrospective chart review was undertaken, but it needs to be specifically stated and the process more defined. The authors should revise accordingly.
Page 2 – Materials and Methods – Paragraph 1 – Lines 92-93 – “The following questionnaires were administered to each patient:” Briefly describe the survey administration process (e.g., who administered it, patient self administration versus investigator/study personnel administration, pencil, and paper format versus on-line survey, etc.). The authors should revise accordingly.
Page 3 – Materials and Methods – Paragraph 3 – Line 113 – “The statistical analysis was performed by using paired Student’s t test…” State whether a commercial statistical software package/program (e.g., SAS, SPSS) was used to manage the data analysis. The authors should revise accordingly.
Page 3 – Table 1 – “Comorbidity” – Define/describe this term. A footnote may be appropriate. The authors should revise accordingly.
Page 4 – Figure 1 – Its significance is unclear and its difficult to interpret. What do the colors represent? Could the figure be deleted since the comorbidities are mentioned in the text. The authors should revise accordingly.
Page 6 – Discussion – Paragraph 1 – Lines 161-162 - “This result differs from other similar studies [24,25].” A brief review of the pertinent findings from these studies (and their limitations) should be included in this section. This will accentuate the differences between those investigations and the current study. The authors should revise accordingly.
Page 6 – Discussion – Paragraph 1 – Lines 163-169 – “While Mabrouk et al., in 2011[26], in line with our results, found no statistically significant differences between before and after the start of hormone therapy. These authors demonstrated that symptom intensity, nodule size, and health-related quality of life remained…” This section is awkwardly worded and confusing. Describe the patient population, therapeutic interventions, the primary and secondary outcome indicators (if any), and the study limitations that could have affected meaningful comparisons to this current study. The authors should revise accordingly.
Page 6 – Discussion - Paragraph 1 – Lines 180-181 – “Another possible consideration concerns the high prevalence of comorbidities in the patients in our sample.” This is an interesting finding and coupled with the association with various phenotypes among women with endometriosis. However, there needs to be a better explanation/description of how these factors influenced the study results. It’s not enough to introduce this concept to the reader without explaining a possible link to the findings. Otherwise, this section of the manuscript doesn’t contribute to the stated study objective and the findings. The authors should revise accordingly.
Page 7 – Discussion – Paragraph 2 – Lines 184 -191- “In recent years, molecular phenotypes are emerging among women with endometriosis that may be related to the presence of a range of diseases, such as inflammatory bowel disease, fibromyalgia, autoimmune diseases, and vulvodynia.” Cite this paragraph with the appropriate reference(s). The authors should revise accordingly.
Page 7 – Conclusions – Paragraph 1 – Lines 200-201 – “Recent papers highlight the possibility of integrating the evaluation of “-omics” data in improving the diagnosis of endometriosis.” Describe “omics” and cite this sentence with the appropriate reference(s). The authors should revise accordingly.
Page 8 – References – Reference 10 has a different citation format than the others. The authors should revise accordingly.
Summary
I commend the authors for undertaking this retrospective review evaluating the effects of medical therapy on the quality of life and sexual function of women with endometriosis. Overall, the manuscript is concise and easy to understand. The authors provide the requisite background for the study and the evidence provided for the reader (albeit limited) supports the study objective. The topic is an interesting one and would be of some interest to the readership who practice in the areas of women’s health and primary care. The findings provide additional insight although questions remain. The conclusion mirrors the study findings. Despite these positive attributes, there were deficiencies noted in the manuscript; mostly errors of omission and the need for further clarification (see full commentary).
The authors briefly address several study limitations that could affect the interpretation and utility of the findings. Such an assessment is appropriate given various study design shortcomings (e.g., sample size and study duration) but additional discussion is warranted. For example, there was no discussion of the shortcomings of the QOL instruments and EMR/chart data reliability and extraction process. One can use study limitations (and the findings themselves) to frame a discussion of specific recommendations involving improved study design attributes (and/or outcomes) for incorporation into similar future investigations (appropriate for the Discussion/Conclusion section). The authors provided recommendations for a multimodal/multidisciplinary approach to manage endometriosis patients given their findings and announced an ongoing prospective study.
The patient care implications of this research can provide import for the reader because it allows for the translation (and interpretation) of the findings into tangible strategies. Given the study design and the limitations, the findings from this study do not allow for making specific recommendations.
I encourage the authors to review the full commentary and make the requisite changes. The deficiencies noted require attention to strengthen the manuscript and its potential utility. I wish the authors continued success in their scholarly endeavors.
Comments on the Quality of English Language
Minor edits due to sentence structure/awkward phraseology
Author Response
Thank you for your review on our paper. I have made the recommended revisions and in the revised version uploaded.

Reviewer 3 Report
Comments and Suggestions for Authors
Manuscript Review: Journal of Personalized Medicine
Title: Does sexual function and quality of life improve after medical 2 therapy in women with endometriosis? A Single-Institution 3 retrospective analysis.
Thank you for the opportunity to review this interesting article about the management of endometriosis.
Abstract: no changes
Introduction:
I think it is important to discuss medical therapy on quality of life, however there is no mention about the role of surgery in correcting anatomic changes and establishing a pathologic diagnosis of endometriosis. The newest research actually discusses that surgery followed by medication management shows the lowest rates of recurrent pain – so there should be at least some mention about how endometriosis can be treated either medically or surgically.
Materials and Methods:
Did any of the patients have surgery during the study period? Was surgery an exclusion criteria? Did some of the patients have previous surgery and were newly starting hormones? Had any of the patients been on hormones in the past – meaning, was this a new initiation of medications?
There needs to be a clear description of what patients met inclusion criteria. The biggest problem of endometriosis research is heterogeneity of the participants which prevents accurate comparison of outcomes. There is a wide range of disease severity, symptoms, and response to various therapy. To lump all patients together without describing how the cohort is selected will lead to significant bias.
What was the actual medical intervention? There is no mention of what the patients were treated with. OCPs, IUD, dienogest, depo provera, progesterone only pills, GnRH analogues? This is essential information. This needs to written in the Methods and then a table needs to be inserted in the Results with the breakdown of medical therapy.
Results:
How do you know that only 9 patients have superficial endometriosis, as this is often not visible on ultrasound or MRI? This must be from the patients who had surgery… but if not all of the patients had surgery, there may be a higher number who had superficial disease. This is why the inclusion criteria must be re-evaluated. I would suggest ONLY including patients who have already had surgery so you can accurately describe the extent of their endometriosis.
Please explain Figure 1 – the image is not easy to understand. Also change spelling of ”Adenomyosis” and “graphical.”
Discussion: please add more references and supporting literature regarding the role of hormone management in endometriosis.
Conclusions: I agree with your comments, but this is the first time you mention individualized care for endometriosis. This is not a conclusion we can gather from the current study; this is an opinion. Your study is designed to determine the QoL outcomes from baseline to 6 months later in patients using hormone therapy. You cannot conclude that a multi-disciplinary approach is needed because you did not study a multi-disciplinary approach in this project.
Comments on the Quality of English Language
Must be improved. Please use the services of an English-language editor.
Lines 41 – 42: grammar – this should be plural
Line 45: incorrect use of colon
Too many short paragraphs that are often only 1-2 sentences long. These should be combined so that each paragraph is articulating a cohesive thought or concept.
Line 76: fix the quotation marks. Typically we don’t need to see the name of the hospital/clinic in the Introduction. This should be described in the Methods.
Line 82 – incorrect space before the period
Multiple incorrect spacings before/after punctuation
Table 1: clean up. What does “NRS.1” mean? “Adenomyosis” is spelled incorrectly. Write out “BMI” as “body mass index” the first time it is used.
Table 3 is written in Italian, please change this to English.
Author Response
Thank you very much for your evaluation of our study. We appreciated the suggestions and have tried to incorporate them into the text in the revised version uploaded.

Round 2
Reviewer 2 Report
Comments and Suggestions for Authors
Reviewer Comments
Page 1 – Introduction – Paragraph 3 – Line 44 – “As a consequence of infiltration of anatomic structures, correlated inflammatory reaction, nociceptive pathway,” This statement is awkward as written. Also, further describe/define “nociceptive pathway”. The authors should revise accordingly.
Page 2 – Introduction – Paragraph 7 – Lines 69-71 – “Furthermore, Caruso and colleagues in 2019 confirmed that this benefit persists in prolonged therapies.[14]” This statement is unclear. Further explain “prolonged therapies”. The authors should revise accordingly.
Page 2 – Materials and Methods – Paragraph 1 – Lines 80-82 - “We evaluated retrospectively patients with diagnosis of endometriosis attending to "Endometriosis and Pelvic Pain" specialized center of the Obstetrics and Gynecology Clinic in Udine, between 2018 and 2020, with a systematic sampling.” Describe who collected the data (investigator versus trained personnel), the records used (EMR versus paper charts) and define the term “systematic sampling”. It’s assumed that a retrospective chart review was undertaken, but it needs to be specifically stated and the process more defined. The authors should revise accordingly. Not addressed
Page 2 – Materials and Methods – Paragraph 1 – Lines 92-93 – “The following questionnaires were administered to each patient:” Briefly describe the survey administration process (e.g., who administered it, patient self administration versus investigator/study personnel administration, pencil, and paper format versus on-line survey, etc.). The authors should revise accordingly.
Page 3 – Materials and Methods – Paragraph 3 – Line 113 – “The statistical analysis was performed by using paired Student’s t test…” State whether a commercial statistical software package/program (e.g., SAS, SPSS) was used to manage the data analysis. The authors should revise accordingly.
Page 3 – Materials and Methods - Paragraph 3 – Lines 127 – 131 – “At this moment all women completed the questionnaire and undergone hormone therapy were informed about the basic goals of the study, and they were briefed about the confidentiality and anonymity. The participation was voluntary. Informed written consent was obtained from each woman, who met the inclusion/exclusion criteria.” This section is out of context to the first sentence of the paragraph. In addition, the first sentence in this section is awkward and unclear. This section refers to the enrollment process and belongs earlier in the description of the methodology. The authors should revise accordingly.
Page 3 – Table 1 – “Comorbidity” – Define/describe this term. A footnote may be appropriate. The authors should revise accordingly.
Page 4 – Figure 1 – Its significance is unclear and its difficult to interpret. What do the colors represent? Could the figure be deleted since the comorbidities are mentioned in the text. The authors should revise accordingly.
Page 6 – Discussion – Paragraph 1 – Lines 161-162 - “This result differs from other similar studies [24,25].” A brief review of the pertinent findings from these studies (and their limitations) should be included in this section. This will accentuate the differences between those investigations and the current study. The authors should revise accordingly.
Page 6 – Discussion – Paragraph 1 – Lines 163-169 – “While Mabrouk et al., in 2011[26], in line with our results, found no statistically significant differences between before and after the start of hormone therapy. These authors demonstrated that symptom intensity, nodule size, and health-related quality of life remained…” This section is awkwardly worded and confusing. Describe the patient population, therapeutic interventions, the primary and secondary outcome indicators (if any), and the study limitations that could have affected meaningful comparisons to this current study. The authors should revise accordingly.
Page 6 – Discussion - Paragraph 1 – Lines 180-181 – “Another possible consideration concerns the high prevalence of comorbidities in the patients in our sample.” This is an interesting finding and coupled with the association with various phenotypes among women with endometriosis. However, there needs to be a better explanation/description of how these factors influenced the study results. It’s not enough to introduce this concept to the reader without explaining a possible link to the findings. Otherwise, this section of the manuscript doesn’t contribute to the stated study objective and the findings. The authors should revise accordingly.
Page 7 – Discussion – Paragraph 2 – Lines 184 -191- “In recent years, molecular phenotypes are emerging among women with endometriosis that may be related to the presence of a range of diseases, such as inflammatory bowel disease, fibromyalgia, autoimmune diseases, and vulvodynia.” Cite this paragraph with the appropriate reference(s). The authors should revise accordingly.
Page 8 – Discussion – Paragraph 6 – Lines 238-243 – “Emerging evidence suggests that neuroinflammation contributes to the pathophysiology of these chronic pain conditions that coexist with endometriosis…” Cite this sentence with the appropriate reference(s). The authors should revise accordingly.
Page 8 – Discussion – Paragraph 7 – Lines 244-246 – “Nociplastic pain should be viewed as an overarching terminology that can be applied to a diverse range of clinical conditions that share common neurophysiological mechanisms, involving various organ systems.” The reference citations ascribed to this sentence do not discuss nociplastic pain. Delete the references (requires renumbering) or choose more suitable citations. The authors should revise acordingly.
Page 8 – Discussion – Paragraphs 6-8 – Lines 238-255 – “Emerging evidence suggests that neuroinflammation contributes to the pathophysiology of these chronic pain conditions that coexist with endometriosis…” These final three paragraphs of the discussion section are problematic (particularly paragraphs 7 and 8).
Paragraph 6 describes the role of neuroinflammation, and the development of chronic pain associated with endometriosis and concomitant conditions. Paragraph 7 describes nociplastic pain and central sensitization (misplled in the text as “sensibilization”) and the co-morbid effects of chronic pain. The last paragraph simply lists CSS diseases, which could probably be included (if appropriate) in paragraph 7. It’s unclear how all 3 paragraphs are linked together and their influence on the study’s findings and/or proposed treatment strategies. Paragraph 5 (Lines 235 -237) may need to come at the end of the discussion along with more tangible management recommendations. The authors should revise accordingly.
Page 8 – Discussion – Paragraph 8 – Lines 251-255 – “Some of these diseases are known as Central Sensitivity Syndromes (CSSs),” Cite this sentence with the appropriate reference(s). The authors should revise accordingly.
Page 7 – Conclusions – Paragraph 1 – Lines 200-201 – “Recent papers highlight the possibility of integrating the evaluation of “-omics” data in improving the diagnosis of endometriosis.” Describe “omics” and cite this sentence with the appropriate reference(s). The authors should revise accordingly.
Page 8 – References – Reference 10 has a different citation format than the others. The authors should revise accordingly.
Summary
I commend the authors for undertaking this retrospective review evaluating the effects of medical therapy on the quality of life and sexual function of women with endometriosis. Overall, the manuscript is concise and easy to understand. The authors provide the requisite background for the study and the evidence provided for the reader (albeit limited) supports the study objective. The topic is an interesting one and would be of some interest to the readership who practice in the areas of women’s health and primary care. The findings provide additional insight although questions remain. The conclusion mirrors the study findings. Despite these positive attributes, there were deficiencies noted in the manuscript; mostly errors of omission and the need for further clarification (see full commentary).
The authors briefly address several study limitations that could affect the interpretation and utility of the findings. Such an assessment is appropriate given various study design shortcomings (e.g., sample size and study duration) but additional discussion is warranted. For example, there was no discussion of the shortcomings of the QOL instruments and EMR/chart data reliability and extraction process. One can use study limitations (and the findings themselves) to frame a discussion of specific recommendations involving improved study design attributes (and/or outcomes) for incorporation into similar future investigations (appropriate for the Discussion/Conclusion section). The authors provided recommendations for a multimodal/multidisciplinary approach to manage endometriosis patients given their findings and announced an ongoing prospective study.
The patient care implications of this research can provide import for the reader because it allows for the translation (and interpretation) of the findings into tangible strategies. Given the study design and the limitations, the findings from this study do not allow for making specific recommendations.
I encourage the authors to review the full commentary and make the requisite changes. The deficiencies noted require attention to strengthen the manuscript and its potential utility. I wish the authors continued success in their scholarly endeavors.
Comments (**Please note the original commentary above was used to provide the review of the revised manuscript)
I commend the authors for undertaking the requisite revisions to the manuscript. Most of the deficiencies were corrected to my satisfaction (those marked in green). Those that were not addressed are marked n purple. Additional comments pertinent to the revised manuscript can be found in dark red. These current issues are significant and require continued author attention. I wish the authors continued success in their scholarly endeavors.
Comments on the Quality of English Language
No major deficiencies regarding the English Language. Some minor problems with sentence structure and phraseology but they do not significantly detract from overall reader understanding.
Author Response
Thanks for these suggestions, we have made the requested changes in the revised file.
Reviewer 3 Report
Comments and Suggestions for Authors
Thank you for making these changes. The methods are more clear, and this is helpful in interpretation of the study. I also think the re-done Discussion and Conclusions is written in a more clear way to address the question.
Can still work on the flow of the Introduction, but overall the manuscript is acceptable with minor spelling and grammar edits.
Comments on the Quality of English Language
Sufficient, no major changes.
Author Response
Thank you for your review, we have made additional changes in the new document.